# The Effects of Residential Built Environment on Supporting Physical Activity Diversity in High-Density Cities: A Case Study in Shenzhen, China

**DOI:** 10.3390/ijerph18136676

**Published:** 2021-06-22

**Authors:** Yuan Gao, Kun Liu, Peiling Zhou, Hongkun Xie

**Affiliations:** 1Harbin Institute of Technology, School of Architecture, Shenzhen 518055, China; altiplano.39@163.com (Y.G.); lindaplzhou@gmail.com (P.Z.); 2Gemdale Group South China Real Estate Company, Shenzhen 518016, China; xiehongkun@gemdale.com

**Keywords:** physical activity diversity, residential built environment, healthy city, high-density city

## Abstract

In high-density cities, physical activity (PA) diversity is an essential indicator of public health and urban vitality, and how to meet the demands of diverse PA in a limited residential built environment is critical for promoting public health. This study selected Shenzhen, China, as a representative case; combined the diversity of PA participants, types, and occurrence times to generate a comprehensive understanding of PA diversity; fully used data from multiple sources to measure and analyze PA diversity and residential built environment; analyzed the relationships between the built environment and PA diversity; and explored the different effects in clustered and sprawled high-density urban forms. PAs in clustered areas were two times more diverse than those in sprawled areas. Accessibility, inclusiveness, and landscape attractiveness of residential built environment jointly improved PA diversity. Clustered areas had significant advantages in supporting PA diversity since they could keep the balance between dense residence and landscape reservation with an accessible and inclusive public space system. The residential built environment with dense street networks, public traffic and service, multi-functional public space system, and attractive landscapes is crucial to improve the diverse PA to achieve more public health outputs in high-density cities. To promote health-oriented urban development, clustered urban form is advocated, and step-forward strategies should be carried out.

## 1. Introduction

In the process of global urbanization, with increasing population densities, public health has gradually become the most urgent urban problem. The role of physical activity (PA) in promoting public health has been confirmed in many studies, such as by preventing chronic diseases and enhancing physical fitness [1], ensuring the physical health of adults and healthy development of children [2,3], guaranteeing residents’ well-being and mental health, and promoting neighborhood communication [4,5,6]. The built environment was proven to influence PA significantly, in which residence and surroundings were believed to be the critical built environment to affect public health [7,8]. In western public health-oriented urban studies, a dense residence with mixed land use, high street connectivity, and large population were indicated to promote PA [9,10,11,12,13]. However, contrary to mainstream research results, high-density cities in Asia have not achieved improved public health outcomes. In the past 30 years, PA in Asia has been declining linearly [14], far below the world average level. Some studies have also found that low residential density is likely to promote PA [15]. This conclusion was also confirmed in many studies on metropolises of western countries [16,17]. Therefore, the consensus on general cities of Europe and America might not be suitable for high-density cities [18]. The discussions on the residential built environment of high-density cities should not only take into account the general built environment indicators but also be rooted in their local socio-economic and urban form characteristics.

In high-density cities, due to the high population density, the PAs are in great and diverse demands [19]. However, the land resources are scarce, and the built environment, especially the residential built environment, cannot provide sufficient space for activities [20]. Thus, improving PA diversity in a limited built environment is an important way for high-density cities to promote public health. Diversity refers to the abundance of things in a certain scope. Diversity is well known in ecology and sociology. It respects the heterogeneity of individuals as a whole and connotes equality and inclusive and continuous symbiosis among individuals. Diversity is the prerequisite and important sign of the vitality and sustainability of things [21,22]. Similarly, the diversity of PA, such as the gathering of a variety of PAs or different groups of people, is to characterize the vitality of a healthy behavior system in a built environment. It should reflect the heterogeneity of PA and the inclusive and symbiotic relationship among various activities [23], especially in high-density cities, since limited public space could not just serve one kind of PAs or a certain group. What the built environment supports is always a complex and mixed social phenomenon. In high-density cities, due to the dense population, it is important to regard PAs as a comprehensive system to meet the multivariate mixed PA demands. Only if we understand the complexity of PA diversity, we could find proper ways to improve the efficiency of a limited built environment on supporting flourishing public healthy life in high-density cities. However, in existing related research, although scholars had noticed the heterogeneity of PA, such as different PA participants [24] and different PA types [25], they were still focused on individual PA, and few studies aggregated PAs into a system to discuss PA diversity.

Back to the basic concept of PA, that is, the activity in which humans use muscles to produce energy consumption, in which there are three basic elements: people, activity forms, and processes to determine the differences of specific PAs in the real world [25,26]. Accordingly, PA diversity should cover at least three kinds of heterogeneity: the diversity of participants, the diversity of PA types, and the diversity of PA processes. The diversity of participants is that various people can perform PA fairly in the same public space, which reflects the fairness of the space to individuals. The diversity of PA types refers to that different activities perform synchronously, which signifies the inclusiveness of the spaces to activities. The diversity of PA processes can be understood as the different occurring times of activities, which indicates the overall continuity of PAs and also reflects the ability of the public space to support activities continuously. These three heterogeneities can collectively represent the fairness, inclusion, and continuous symbiosis among various PAs and fully describe the abundance of healthy behaviors in high-density cities (Figure 1). Based on the above understanding, this research aimed to represent the levels of PAs from a diversity perspective in high-density cities, to explore the relationship between the residential built environment and the PA diversity, and to put forward the planning and design strategies for the effective promotion on public health.

## 2. Methods

### 2.1. Research Area

Shenzhen is located in the Pearl River Delta region of southern China, close to Hong Kong. It is the first city to open up as part of China’s reform policy and is one of the core cities in the Guangdong-Hong Kong-Macao Greater Bay Area. Statistics from 2020 show that Shenzhen has an administrative area of 1997.47 km^2^, including 975.5 km^2^ for construction, with a service population of 20 million people. It is the city with the smallest area and largest population among all metropolises in China.

Shenzhen was established as a city in 1980, starting from the Shenzhen Special Economic Zone (SEZ), which was approximately 327.5 km^2^ and commonly referred to as “Inside Shenzhen”. The SEZ carried out top-down urban construction as part of the overall urban planning, forming a sparse and orderly banded-cluster structure. Each cluster is with an appropriate scale (around 10 km^2^), has mixed land-use and high-density street networks, and is bordered by natural elements such as rivers and hills. Meanwhile, around the SEZ, the “Outside Shenzhen”, covering more than 1600 km^2^, was built up in a disordered bottom-up way, with rapid urban sprawl along external traffic corridors. Due to the lack of forwarding planning management, the density of street networks and public services are much poorer than those inside the SEZ. As a result, two distinct urban forms were shaped: clustered and sprawled [27,28]. As shown in Figure 2, these two kinds of urban forms provided two typical high-density built environment samples for this research.

### 2.2. Research Design

Based on the comprehensive understanding of PA diversity, this research defined PA diversity as integrated characteristics of PA richness in terms of PA participants, PA types, and PA process. Considering the heterogeneity of Shenzhen’s population was more based on social-economic attributes than on age or ethnicity since Shenzhen was a free-market-oriented young immigrant city, income was selected to present the PA participants’ heterogeneity.

In this research, the residential built environment was defined as the neighborhoods where PAs occurred, including not only the residence where PAs started or ended but also the surrounding built environment where PA passed. Built environment indicators that affect PA diversity in existing research were summarized into four parts: accessibility, utility, inclusiveness, and landscape attractiveness. First of all, accessibility was commonly believed to support all kinds of PAs [29,30], in which access to service facilities and public transit positively supports walking and jogging [31,32,33], and the density of pedestrians and intersections can promote the presence of PAs [11,12] and inspire PA in multiple social classes [34]. Secondly, the utilitarian aspects of the built environment, such as mixed land use and population density, have also been indicated to positively support PA types and different PA participants [35,36,37,38,39,40,41,42]. Further, studies have shown that the built environment also attracts more PAs by providing inclusive places and increasing sports facilities, such as playgrounds and greenways [43,44,45,46]. Meanwhile, the landscape attractiveness of the built environment, such as places with scenic spots and a high amount of green coverage, can continuously attract more kinds of PA effectively [44,47]. In summary, accessibility is a common feature of the built environment that supports almost all kinds of PAs, and the utility of the built environment provides the opportunity for PA occurrence. The inclusiveness of the place serves to promote the co-occurrence of multiple types of PA, and the landscape could attract various people and activities. We hypothesized that the residential built environment supported PA diversity by four characteristics: accessibility for people to perform PA, a utilitarian environment to provide an opportunity for different kinds of PAs, inclusiveness to support the co-occurrence of various PAs, and landscape attractiveness to gather various PAs. At the same time, we hypothesized that the effects of the residential built environment to support PA diversity were different between clustered and sprawled high-density urban forms.

### 2.3. Data Resources

#### 2.3.1. PA Data

Compared with other types of PA, walking, jogging, and cycling are the most common PAs in cities [48]. These three activities do not rely on professional sports venues and equipment, mostly occur in urban public spaces, and are closely related to the built environment. This article selects walking, jogging, and cycling as the PA research objects and collects PA data through the PA tracking application Codoon, which is one of China’s most popular self-tracking applications. With Codoon, people can track PA information (including the type of activity, time, speed, duration, and routes) and upload, share, and compare their workouts on social network platforms, which forms a natural VGI data pool. By the end of 2016, Codoon was the most commonly used PA tracking application in China, and the number of users in Shenzhen ranked among the top five Chinese cities [48]. Besides, the PA data from Codoon cover 24 h, which can ensure comprehensive coverage of PA occurrence time. The majority of Codoon’s users are young and middle-aged [49]. Similarly, the average age of the population in Shenzhen is relatively young, with a median age of 31.95 [50]. The structure of Codoon’s users is similar to that of the population in Shenzhen, so the PA data from Codoon are representative in this research.

After counting the number of Codoon PA data in Shenzhen each month of 2015, it was found that the number of PAs was highest in spring and summer. On this basis, April and July were selected as typical time periods. The preliminary research also found that the number and the spatial distribution of daily PA data were relatively stable, with a difference between weekdays and weekends. Therefore, this study selected 8 days to collect daily full sample data, including two weekdays and two weekends in April and July.

To focus on the residential built environment, those samples that started or ended on residential land were chosen. All the PA data were divided into three types: the ones that started in residential land and ended in residential land, and those that started and ended in the same residential land. Those that started and ended in different residential areas were eliminated since it was hard to judge the living address of the participants. The housing price data from Home Link, which was the most popular housing transaction platform, were used to estimate the income of participants. There were 735 data, including 194 walking, 486 jogging, and 55 cycling data.

#### 2.3.2. The Residential Built Environment Data

The residential built environment data included: the Shenzhen Land-Use Survey (2014) from Shenzhen Land Use Master Plan (2006–2020), in which land was classified into nine types (residential land, commercial land, government and institutional land, industrial land, warehouse land, street land, infrastructure land, parklands, and other lands); street network data from the Open Street Map, including five types (motorway, primary, secondary, branch, and others); bus stop data from Baidu Map Point of Interest (2012); greenway networks from the Shenzhen Greenway Map (2013); sports facilities, restaurant data, service facilities (libraries, museums, cultural centers, hospitals, and banks), and scenic spots from the Scott Map (2016).

### 2.4. Measurements

#### 2.4.1. Measurements Unit

In the field of geography, grid networks are often used as spatial units to analyze natural and social phenomena. The size of the grid used in general research is between 500 m and 1000 m [51,52]. In the study of high-density cities or densely populated areas, a grid with a side length of 500 m is often used as a calculation unit [53,54,55]. To measure the characteristics of PA diversity and built environment, a 500 m × 500 m grid [56,57,58] was used as the calculated unit [59,60]. By using the Fishnet Tool in ArcGIS, the land in Shenzhen was divided into grids; those without PAs were excluded, which left 2049 grids. Since the PA samples started or ended in residential land, the scope of the grids with PAs was defined as a residential built environment.

#### 2.4.2. Dependent Variable

Based on the integrative definition, PA diversity is a comprehensive indicator that includes PA participant diversity (PAPD), PA type diversity (PATD), and PA occurrence time diversity (PAOD). Shannon’s Diversity Index was used to measure diversity [61] (Table 1) since it was often used in the calculation of land use mixture [13,62] and PA diversity [51]. The index varies between 0 and 1 (0 for maximum specialization, 1 for maximum diversification). PAPD referred to the abundance of the participants’ income levels [63] (Equation (1)); PATD was the richness of different activities, including walking, jogging, and cycling (Equation (2)); PAOD was used to measure the diverse occurrence times from morning to night (Equation (3)).

The income level of each participant was estimated by the housing price to income ratio (HPIR), which was 27.7 in 2015 [64]. The Joins tool in ArcGIS was used to assign housing price information to the residential lands of PA origin destinations to obtain the price per square meter of commercial housing where each participant lived, and then the annual family income was inferred by Equation (4). The income was divided into four categories: high-income level, middle-income level, low-income level, and lower-income level. (Table 2). Affordable housing and urban villages had no housing prices since they could not be traded. Considering relevant policies [65], people who lived in affordable housing were classified as low-income level. Additionally, because lower-income groups mostly live in urban villages [66], those participants from or to urban villages were defined as lower-income level.

This is example 1 of an equation:(4)n=m×yHPIRwhere *HPIR* was the housing price to income ratio; *m* was the price per square meter of commercial housing; *y* was the average residential area, and *n* was the annual family income.

PA diversity was obtained by reducing the dimensions of PAPD, PATD, and PAOD using principal component analysis in SPSS [69]. The KMO was 0.812, which indicated the data were suitable for factor analysis, and Bartlett’s Test of Sphericity was 0.000 to provide a reasonable basis for factor analysis. Table 3 showed that the result of principal component analysis had a high degree of interpretation.

#### 2.4.3. Independent Variables

According to the hypothesis, the residential built environment was measured from four aspects: accessibility, utility, inclusiveness, and landscape attractiveness. Accessibility included the street network density [11,12,34,43], bus stop station density [31], and public service facility density [32,33], which represented the accessibility of potential destinations in a city. Utility included the population density [40,70,71], restaurant density [72], and land use mixture [42,43,44,45,46,47,71], which directly reflected the utility of built environment. Inclusiveness included sports facility density [73] and the greenway network density [41]. Greenway is a linear public space that facilitates citizens moving close to natural resources such as rivers, lakes, and mountains [52], and it is generally designed for popular PAs such as walking, jogging, and cycling [53]. Sports facilities are also built to support diverse types of PAs. Landscape attractiveness included the green land ratio [47] and scenic spot density [74]. Indicators and measures are shown in Table 4.

#### 2.4.4. Covariates

Considering that PAs were affected by seasons and holidays, two control variables were selected: air temperature and weekend status. The highest daily average temperature in April was 28 °C, and the lowest daily average temperature in July was 28 °C [75], this study selected 28 °C as a cut-off point: 1 for 28 °C and below, 0 for above 28 °C. Weekend status was measured by dichotomous variables, 1 for weekends and 0 for weekdays. Since PA data obtained from the internet did not contain individual attributes, personal factors were not discussed in the models.

### 2.5. Data Merging and Statistical Analysis

The PA and residential built environment data were overlaid in ArcGIS 10 with the Shenzhen local coordinate system to form the database. The Join tool in ArcGIS was used to link each feature value of the residential built environment with each 500 m grid corresponding to the PA diversity according to the label of each 500 m grid so that each grid had both PA diversity and built environment attribute. Each variable in each grid was averaged. SPSS was used for statistical and regression analyses to establish multivariate linear regression models to explore the association between PA diversity and the residential built environment. Three hypothetical models were made for the overall, clustered, and sprawled urban areas separately (Table 5). The collinearity diagnostics among the independent and control variables were performed before modeling, and the results showed no strong associations among the variables (VIF < 10). To make valid inferences from the regression, this paper tested the residuals of the regression, and the residuals were normally distributed.

## 3. Results

### 3.1. Descriptive Characteristics

#### 3.1.1. PA Diversity and Spatial Distribution

Among the 2049 spatial units, the mean value of PA diversity was 1.000. The comparison of PA diversity in clustered and sprawled areas indicated that the PAs in clustered areas were two times more diverse than those in sprawled areas (Table 6). Figure 3 shows that PA diversity was unevenly distributed in Shenzhen. The most diverse PAs were widely spread and exhibited line-like aggregation patterns in clustered areas along seasides, rivers, within large parks, and downtown residential areas. A few units with diverse PAs in the sprawled area presented a spot-like aggregation pattern in the downtowns.

#### 3.1.2. The Characteristics of the Residential Built Environment

The statistical analysis showed the similarities and differences in the residential built environment in clustered and sprawled areas. As shown in Table 7, both areas had similar populations and land use mixtures. Because of the strict ecological control system and top-down overall greenway network constructions, the density of green resources and greenways were at the same level. However, scenic spots in the clustered area were dense and mostly accompanied by greenways (Figure 4). The street network, bus station, and public service in the clustered area were much denser to reflect a more accessible environment. Besides, the clustered area had much richer restaurants and sports facilities than in the sprawled area (Figure 5). In the clustered area, each cluster was organized in a network with complete facilities to form a self-sufficient unit, and clusters were divided by rivers, hills, and green belts to maintain many scenic spots. In the sprawled area, construction lands continuously expanded, and there was a lack of overall transportation and facility planning, which resulted in poor accessibility and insufficient facilities.

### 3.2. Statistical Modelling Results

Three multivariate linear regression models were developed separately for the overall, clustered, and sprawled areas to explain and compare the relations between the residential built environment and PA diversity. The adjusted R squared presents the goodness of fit in the three models.

Table 8 shows the different results among the three models. In model 1, accessibility, inclusiveness, and landscape attractiveness were all positively related to PA diversity, and utility partly had negative effects. Comparing the results of models 2 and 3, the accessibility and utility of the residential built environment were closely associated with PA diversity in the sprawled area and slightly affected PA diversity in the clustered area. While inclusiveness and landscape attractiveness were positively related to PA diversity in the clustered area, they contributed little to diversity in the sprawled area. In detail, PA diversity in the clustered area was positively associated with the green space ratio (95% CI 0.010, 0.587; *p* value = 0.043), greenway network density (95% CI 0.104, 0.181; *p* value = 0.000), sports facility density (95% CI 0.002, 0.010; *p* value = 0.008), and scenic spot density (95% CI 0.001, 0.009; *p* value = 0.010) and was negatively associated with the land use mixture (95% CI −0.997, −0.211; *p* value = 0.043). PA diversity in the sprawled area was positively associated with street network density (95% CI 0.017, 0.033; *p* value = 0.000), bus station density (95% CI 0.000, 0.002; *p* value = 0.050), service facility density (95% CI 0.003, 0.006; *p* value = 0.000), and greenway network density (95% CI 0.002, 0.057; *p* value = 0.032) and was negatively associated with population density (95% CI −1.553, −0.075; *p* value = 0.031).

## 4. Discussion

### 4.1. The Important Role of Accessibility

The results show that the accessibility of the residential built environment is positively associated with PA diversity in the overall and sprawled areas, which indicates the significance of street network density, bus stop density, and service facility density for supporting diverse and vital physical activities. However, this relationship was not significant in the clustered area.

As shown in Table 7, the accessibility of the residential built environment in the clustered area was much higher than that in the sprawled area. Taking 0.5 as the dividing point, the PA diversity was divided into high and low parts. The comparison of the accessibility between high and low parts indicated that there was no significant difference in the clustered area, while in the sprawled area, the accessibility of the high part was twice that of the low part. Within the high parts, the accessibility in the clustered area was the same as that in the sprawled area. Because of the generally high accessibility in the clustered area, there were no significant relations between accessibility and PA diversity, so the irrelevance result did not mean that accessibility was not important in the clustered area. On the contrary, an accessible built environment with a dense street network, public traffic, and public service played necessary roles to activate various physical activities.

### 4.2. The Negative Effects of Utility on PA Diversity

The results showed that the utility of the residential built environment was negatively correlated with PA diversity, in which land use mixtures reduced PA diversity in the clustered area, and population density had negative effects on PA diversity in the sprawled area. Preliminary research has shown that a high land-use mixture and dense restaurants could enhance individual travel and induce individual PA [44]. However, the residential areas with the dense and mixed-use built environment are often located in downtowns, where it is difficult to provide enough space to gather for various types of PA. To date, dense and mixed-use residential blocks had failed to provide spaces for various PAs to achieve healthy vitality.

### 4.3. The Co-Occurrence Effects of Inclusiveness on PAs

In both clustered and sprawled areas, the greenway network density was found to play a significant role in promoting PA diversity. This also reflected that the greenways effectively support different kinds of activities since they were specially designed for various PAs. Since 2010, a total of 2448 km of greenways was built, forming a large and homogeneous greenway network throughout Shenzhen (Figure 3), which provided almost the same opportunities to attract various PAs in both clustered and sprawled areas and achieved co-occurrence effects to support diverse PAs. In addition, sports facility density was found to promote PA diversity in the clustered area, while it had no effect in the sprawled area. As a professional public service for PA, sports facilities are undoubtedly capable of attracting diverse PAs. In the clustered area, sports facilities and the greenway network together formed a system to support diverse PAs, while in the sprawled area, due to weak planning and the backward stage of development, the number of sports facilities was in short supply, and they were not integrated with the greenway network, which failed to promote PA diversity.

### 4.4. The Aggregation Effects of Landscape Attractiveness on PA Diversity

In models 1 and 2, the green space ratio and scenic spot density were positively correlated with PA diversity. Overlapping the green lands, scenic spots, and PA diversity (Figure 4), it was clear that places with rich green and scenic spots corresponded to high PA diversity, which showed clear aggregation effects on diverse PAs. Unlike greenways and sports facilities, which functionally support various types of PA, places with good landscapes provide rich visual and mental experiences, so they have become popular destinations that can aggregate diverse PAs. In the sprawled area, green lands were as rich as those in the clustered area, but because of the marginal spatial distribution and poor accessibility, they were difficult to reach and use. Meanwhile, the sparse scenic spots of the residential built environment resulted in weak attractiveness, thus failing to gather diverse PAs.

### 4.5. The Joint Effects of the Built Environment on PA Diversity

Based on the above discussions, we found because of the overall PA-friendly travel environment, the evenly distributed multi-functional public space systems, and rich landscape, the clustered area provided sufficient accessible, inclusive, and attractive places for various PAs to achieve high PA diversity. Meanwhile, in sprawled areas, although the greenway networks and green spaces were as rich as those in clustered areas, the poor accessibility, sparse, and discrete public space system still cause to attract diverse PAs. These results indicate the joint effects among accessibility, inclusiveness, and landscape attractiveness of built environment on PA diversity, in which accessibility is the fundamental and necessary condition to provide possibilities for people to go outside and perform various PAs, inclusive public systems such as greenways with dense sports facilities can functionally support the co-occurrence of various PAs to improve diversity, and good landscapes further trigger rich activity experiences to strengthen the aggregation of various PAs, thus resulting in increased PA diversity.

### 4.6. Advantages of Clustered Areas for Promoting Healthy and Vital Urban Life in High-Density Cities

An accessible, inclusive, and attractive residential built environment is important to improve PA diversity and promote the vitality of a healthy city. However, in high-density cities, crowded and expanded urban construction easily results in poor accessibility, scarce public space, and less attractive landscapes. In this research, compared with sprawled areas, clustered areas successfully supported PA diversity, which indicated the advantages of clustered urban form. In SEZ, the city was divided into several clusters with appropriate size by natural reservations such as rivers and hills, and each cluster was built with a dense street network and services with systemic public space to form an integrally organic and inner compact living unit, then achieved accessible, inclusive, and attractive built environment on supporting various PAs. In high-density cities, it is crucial to maintain the balance between dense residence and landscape reservation and build up an accessible and inclusive public space system such as a greenway network for promoting healthy and vital urban life.

### 4.7. Limitations

This study fully utilized big data from multiple sources, including traditional statistics, PA VGI, and open-source data such as POIs and housing prices. Because it was difficult to guarantee the same acquisition times for multi-source data, the combined analysis of data might lead to errors and affect the accuracy of the results. Meanwhile, due to the limitation of data collection methods, providers of PA VGI data were probably young and modern people; accordingly, the descriptions and measures of PA might not be generalized. Furthermore, due to defects in PA VGI data, the individual characteristics of PA participants could not be included in the models, resulting in the insufficient explanatory ability of the study. An offline social survey of Codoon users should be added for collecting more demographic information to improve the accuracy of the model in future studies.

## 5. Conclusions

This study identifies the importance of PA diversity in public health research in high-density cities. PA diversity was defined as a comprehensive concept that contains PA participant diversity, PA type diversity, and PA occurrence time diversity to indicate the fairness, inclusive, and continuous symbiosis among various PAs and the abundance of healthy behaviors. The systematic discussion of PA diversity provides a new way for research on PAs in the field of urban planning and design, especially in the context of high-density urbanization all over the world.

The correlation analysis revealed the joint relationships between the residential built environment and PA diversity in high-density cities. Accessibility is the fundamental condition to activate individual PAs, inclusiveness supports the co-occurrence of various PAs to consolidate PA diversity, and landscape attractiveness furtherly aggregates multiple PAs.

The research also found that the PA diversity in the clustered area was higher than that in the sprawled area. This result occurred because the clustered urban form structurally maintained the balance between dense residence and landscape reservation with an accessible and inclusive public space system. To promote health-oriented urban planning and development in high-density cities, the clustered urban structure should be advocated, and step-forward strategies should be carried out: first, to promote the overall accessibility of the residential built environment to promote PA; second, to build an inclusive public space system for the co-occurrence of various types of PAs; and further, to improve the attractiveness of landscapes to aggregate PAs for higher diversity. In summary, creating an accessible, inclusive, and attractive residential built environment is a crucial way to synchronously support various PAs to improve the vitality of public health in high-density cities.

## Figures and Tables

**Figure 1 ijerph-18-06676-f001:**
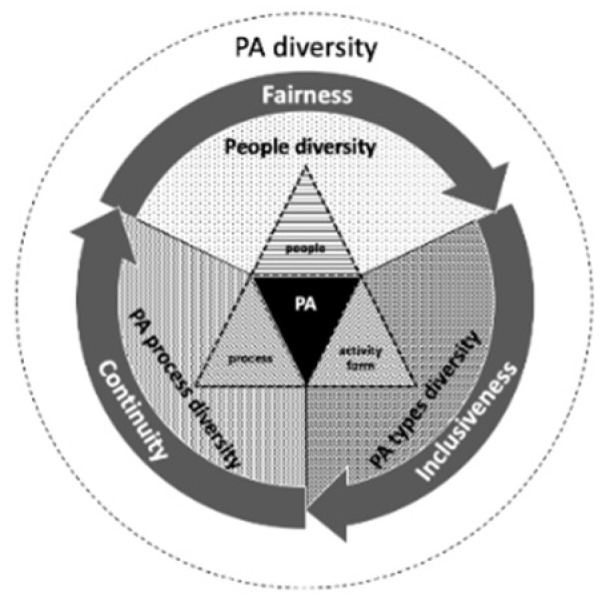
The conceptual model of PA diversity.

**Figure 2 ijerph-18-06676-f002:**
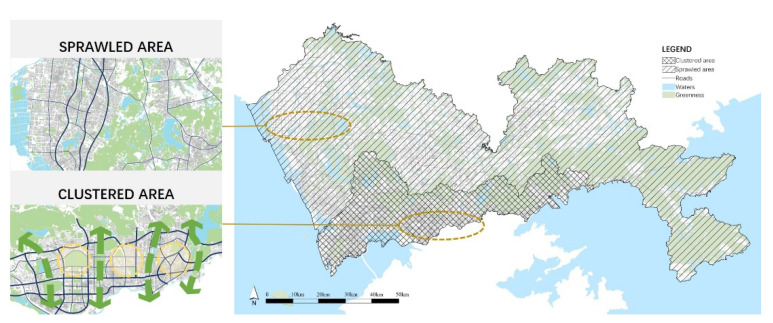
Schematic diagram of Shenzhen.

**Figure 3 ijerph-18-06676-f003:**
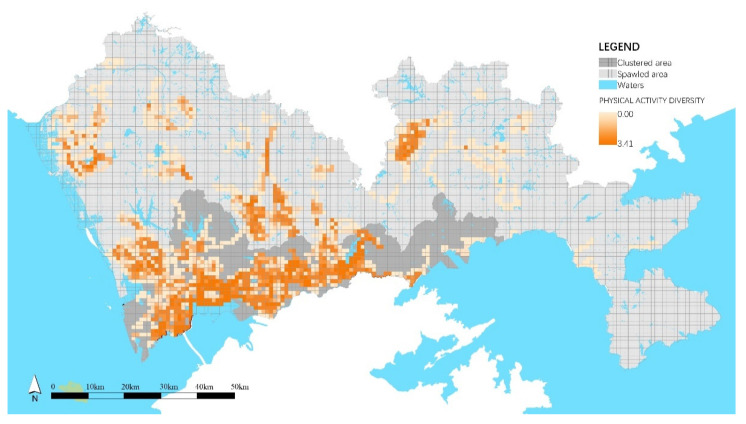
Spatial distribution of PA diversity.

**Figure 4 ijerph-18-06676-f004:**
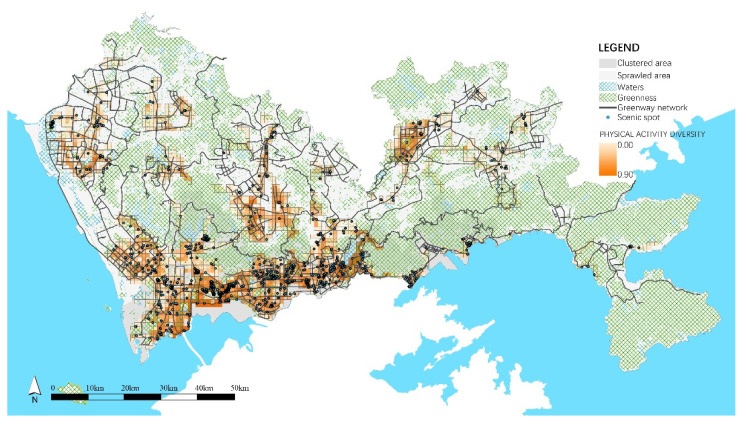
Spatial distribution of greenway network and scenic spots.

**Figure 5 ijerph-18-06676-f005:**
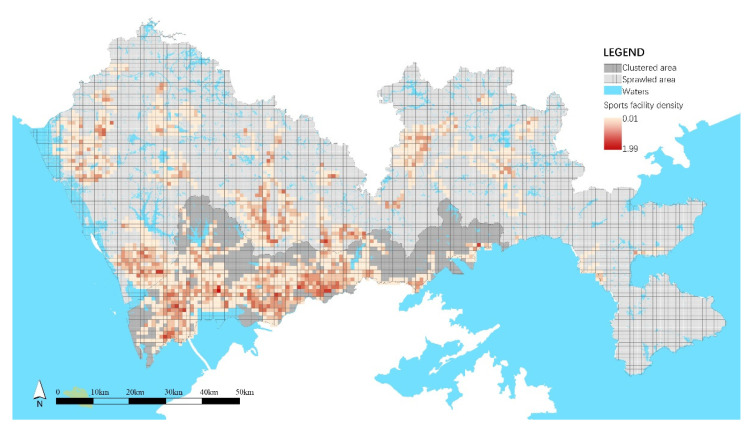
Spatial distribution of sports facility density.

**Table 1 ijerph-18-06676-t001:** Measures of the three dimensions of PA diversity.

The Dimension of Diversity	Measure
PAPD	$PAPD=\frac{\left\{-\sum {}_{k_{1}}\left[\left(p_{1}\right)\left(\ln p_{1}\right)\right]\right\}}{\ln k_{1}}$ (1) where *k*_1_ = number of social class types, according to income (*k* = 4: lower-income class, low-income class, middle-income class, high-income class); *p*_1_ = the proportion of the PA distance of each social class to the PA distance of all people in each 500 m grid.
PATD	$PATD=\frac{\left\{-\sum {}_{k_{2}}\left[\left(p_{2}\right)\left(\ln p_{2}\right)\right]\right\}}{\ln k_{2}}$ (2) where *k*_2_ = number of physical activity types (*k* = 3: cycling, walking, jogging); *p*_2_ = the proportion of each type of activity distance to all activity distances in each 500 m grid.
PAOD	$PAOD=\frac{\left\{-\sum {}_{k_{3}}\left[\left(p_{3}\right)\left(\ln p_{3}\right)\right]\right\}}{\ln k_{3}}$ (3) where *k*_3_ = number of physical activity times (*k* = 4: from 00:00 to 06:00, from 06:00 to 12:00, from 12:00 to 18:00, from 18:00 to 24:00); *p*_3_ = the proportion of the PA distance of each time period to the PA distance of all time periods in each 500 m^2^ grid.

**Table 2 ijerph-18-06676-t002:** The classification standard of income level.

Classification	*HPIR*	*y*	*n*
high-income level	27.7 [64]	80 m^2^ [67]	≥18,773 USD [63,68]
middle-income level		80 m^2^ [67]	18,773–7509 USD [63,68]
low-income level		60 m^2^ [67]	7509–3755 USD [63,68]
lower-income level		60 m^2^ [67]	≤3755 USD [63,68]

**Table 3 ijerph-18-06676-t003:** Total variance explained in the principal component analysis.

	Initial Eigenvalues			Extraction Sums of Squared Loadings		
Component	Total	%Variance	Cumulative%	Total	%Variance	Cumulative%
Factor 1	2.339	77.981	77.981	2.339	77.981	77.981
Factor 2	0.389	12.956	90.937	-	-	-
Factor 3	0.272	9.063	100.00	-	-	-

**Table 4 ijerph-18-06676-t004:** Variables and measures of the residential built environment.

Variable	Measure
**Accessibility**	
Street network density	The total length of streets per unit area
Bus station density	Number of bus stops per unit area
Service facility density	Number of service facilities per unit area
**Utility**	
Population density	The ratio of residential building area to total building area per unit area
Restaurant density	Number of service facilities per unit area
Land use mixture	Shannon’s Diversity Index was used to calculate the land-use mixture, which varied between 0 and 1 (0 for maximum specialization, 1 for maximum diversification) $Land\;use\;mixture=\frac{\left\{-\sum {}_{k_{4}}\left[\left(p_{4}\right)\left(\ln p_{4}\right)\right]\right\}}{\ln k_{4}}$ where *k*_4_ = number of land use types (*k* = 9: residential land, commercial land, government and institutional land, industrial land, warehouse land, street land, infrastructure land, parklands, and other lands); *p*_4_ = the proportion of each type of land within the 500 m grid.
**Inclusiveness**	
Greenway network density	The total length of the greenways per unit area
Sports facility density	Number of sports facilities per unit area
**Landscape attractiveness**	
Green space ratio	The proportion of green space per unit area
Scenic spot density	Number of scenic spots per unit area

**Table 5 ijerph-18-06676-t005:** The hypothetical models.

Variable	Model 1	Model 2	Model 3
	Overall	Clustered Area	Sprawled Area
**Accessibility**			
Street network density	Positive	Positive	Positive
Bus station density	Positive	Positive	Positive
Service facility density	Positive	Positive	Positive
**Utility**			
Population density	Positive	Positive	Positive
The proportion of commercial land	Positive	Positive	Positive
Land use mixture	Positive	Positive	Positive
**Inclusiveness**			
Greenway network density	Positive	Positive	Positive
Sports facility density	Positive	Positive	Positive
**Landscape attractiveness**			
Green space ratio	Positive	Positive	Positive
Scenic spot density	Positive	Positive	Positive
Temperature ≤ 28 °C	Positive	Positive	Positive
Weekend	Positive	Positive	Positive

**Table 6 ijerph-18-06676-t006:** The description of PA diversity.

	Mean	Maximum	Minimum	Standard Deviation
Clustered area	1.5162	3.4100	0.0700	1.0351
Sprawled area	0.6767	3.2000	0.0000	0.8274
Overall	1.0000	3.4100	0.0000	1.0000

**Table 7 ijerph-18-06676-t007:** Description of the characteristics of the residential built environment.

Variables	Mean		
	Overall	Clustered Area	Sprawled Area
**Accessibility**			
Street network density (km/km^2^)	8.484	12.220	6.141
Bus station density (Pcs/km^2^)	54.330	69.389	44.901
Service facility density (Pcs/km^2^)	32.015	39.849	27.110
**Utility**			
Population density (5900 person/km^2^)	0.071	0.066	0.073
Restaurants density (Pcs/km^2^)	0.033	0.045	0.026
Land use mixture	0.454	0.468	0.446
**Inclusiveness**			
Greenway network density (km/km^2^)	1.729	1.784	1.695
Sports facility density (Pcs/km^2^)	8.562	12.068	6.367
**Landscape attractiveness**			
Green space ratio	0.208	0.206	0.209
Scenic spot density	2.308	4.313	1.053

**Table 8 ijerph-18-06676-t008:** Regression models and results.

Variables	Mean										
	Overall			Clustered Area				Sprawled Area			
	B	Beta	Sig.	B		Beta	Sig.	B		Beta	Sig.
**Accessibility**											
Street network density	0.025	0.183	0.000	0.005		0.038	0.242	0.025		0.178	0.000
Bus station density	0.001	0.067	0.003	0.001		0.051	0.156	0.001		0.060	0.050
Service facility density	0.003	0.117	0.000	0.001		0.038	0.409	0.004		0.206	0.000
**Utility**											
Population density	−0.575	−0.044	0.075	−0.562		−0.041	0.291	−0.814		−0.077	0.031
Restaurants density	−0.001	−0.033	0.254	0.000		0.037	0.411	−0.001		−0.022	0.613
Land use mixture	−0.235	−0.043	0.050	−0.604		−0.105	0.003	−0.023		0.005	0.868
**Inclusiveness**											
Greenway network density	0.065	0.103	0.000	0.142		0.221	0.000	0.030		0.057	0.032
Sports facility density	0.005	0.073	0.003	0.006		0.103	0.008	−0.001		−0.010	0.774
**Landscape** **Attractiveness**											
Green space ratio	0.260	0.067	0.003	0.297		0.076	0.043	0.057		0.017	0.578
Scenic spot density	0.008	0.084	0.000	0.005		0.076	0.010	0.006		0.024	0.331
**Weekend**	0.525	0.263	0.000	0.794		0.356	0.000	0.314		0.190	0.000
Temperature ≤28 °C	0.529	0.258	0.000	0.595		0.251	0.000	0.443		0.268	0.000
Constant	−1.015		0.000	−0.717		-	0.000	−1.007		-	0.000
**Adjusted R square**	0.371				0.366				0.291		

## Data Availability

Publicly available datasets were analyzed in this study. The PA data in this study were collected from Codoon, it is one of China’s most popular self-tracking applications. Its data is publicly available. Users of Codoon can view the activity information of other users by logging in to the official website and registering an account for free.

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
