# Peer review of "The Effects of Residential Built Environment on Supporting Physical Activity Diversity in High-Density Cities: A Case Study in Shenzhen, China"

_ijerph, 2021, doi:10.3390/ijerph18136676_

Round 1
Reviewer 1 Report
The paper is quite clear & straight-forward. Aims are clearly outlined. Authors also clearly explained each subsection to make it easy to understand. However, some sections of the Methodology were technical.
Line 50: "What is diversity?" can be omitted to just go ahead to define it.
Line 54: "diversity of PA" is this a term coined by the authors or it's a known term?
Line 79: what does heading "Subsection" mean?
Line 82: can authors find a much recent statistics report on Shenzhen?
Line 334: greenway could be defined after its introduction in line 223.
Author Response
Dear Reviwer
please see the attachment

Reviewer 2 Report
This is an interesting report. I have the following comments for the authors.
Specific comments:
- The general health benefits of physical activity, be it a short-bout or long-term should be more clearly stated in the introduction section. For example, exercise has been linked to increased blood flow to the brain and neurotransmitter levels, enhanced plasticity and better focus, attention and information processing in typically-developing children and children with attention-deficit/hyperactivity disorder (citation: pubmed.ncbi.nlm.nih.gov/28917364).
- "PA diversity in the sprawled area was positively associated with street network density" - please provide the appropriate statistic, 95% CI and p-value to support this assertion.
- "Is accessibility not important in clustered areas?" - please rephrase this as a statement instead of a question.
- "Three characteristics are interrelated: the former is a necessary condition for the latter, and the latter is a sufficient condition for the former" - how did the authors conclude this based on the present analysis?
- Please avoid using exaggerated words and phrases, such as "PA diversity is the comprehensive embodiment of the vitality of a healthy city", etc. This kind of sentence is acceptable for an editorial or a perspective article but not for this type of article (e.g., original research or systematic reviews).
- The underlying data should be made publicly available. If this was not possible, please provide a reason why.
Author Response
Dear Reviwer
Pleaase see the attachment

Reviewer 3 Report
I have read with great interest the manuscript. I have found it very well designed and structured. The data is well presented and the methodology is very appropriate. The authors have proceeded with great rigour in their analysis. However, I have found that the discussion and the contextualization of the problem (the state of the art) is very concentrated in Chinese literature. Nevertheless, some of those references and published in international journals (non-chinese ones). Thus, it would be important for the authors to check among other literature (non-chinese) if the problematics addressed are also an issue, and try to contextualize their work on those cases. I believe with this, the authors will gain greater international relevance and will not enclose their work only in the Chinese context. Bringing that into the introduction and state of are and later discuss such issues in comparison to the author's analysis, in the conclusions, will be very useful.Author Response
Dear Reviewer
Please see the attachment

Reviewer 4 Report
This study investigates the relationship between PA diversity, defined through 3 dimensions, and the built environment characteristics in high-density neighbourhoods in Shenzhen, China. The manuscript contributes to the knowledge on built environment correlates of PA diversity by studying PA as an aggregate variable and by focusing on an Eastern, high-density city, as compared to its western counterparts. However, there are many unclear parts in the manuscript, which need clarification before the paper can be publishable.
Abstract
- “the compatibility and utilization efficiency of limited residential built environment”; “progressive effects”- These terms/sentences are not clear. Please clarify.
- “Accessible, inclusive, and attractive”- these are too broad terms-need specifics- maybe give examples, i.e. xyz..
- The keywords should be reviewed- I do not see how “progressive effects” can be a keyword- what does this mean?
Introduction
- Line 45: “In high-density cities, due to the high population density, the PAs are in great and diverse demands.” Reference?
- Line 48: “compatibility and utilization efficiency”- what does this mean? Too vague.
- Line 50: “Diversity refers to the abundance of things”- not agree on this definition-i.e. some land uses might be in abundance, but if they are not equally distributed, then diversity in specific areas might be non-existent...
- Line 54: “Diversity of PA”- needs to be clarified and exemplified- what does this mean exactly?
- Line 58-59: “few studies aggregated PAs into a system to discuss PA diversity.”- why is this important?
- Line 72: “this research aims to comprehensively present the abundance of PAs in high-density”- why? why not say: to represent the levels of PAs in high-density cities? Is this not more objective? otherwise, is it not biased?
- Line 89: “Each cluster has mixed land-use and street networks” - what does "mixed street network" mean? what type of street network do clusters have? how do they differentiate or compare?
- Line 95: “clustered and sprawled” -how are these defined/measured?
- Lines 107-109: Definition of residential areas- this is not clear and controversial- so how was residential built environment defined? how does it differ from a commercial area, for example? This needs to be clarified.
- Line 115: Please clarify “utilitarian aspect of the built environment”.
- Line 118: “providing inclusive places”- what does this mean? Please clarify.
- Figure 3: no need to include this image-this is just a brief summary of the 4 categories and not a research framework.
Methodology
- Line 153: “PAs was highest in spring and summer”- why not select 1 month with high PA and 1 month with lower PA?
- Line 155: “with a difference between weekdays and weekends.” -was there a significance difference?
- Line 158: not clear how housing price data was matched with participant data? do the authors have residential addresses of participants? in which level- postcode, street address, etc. ?
- Line 165: “street land, infrastructure land,”- what is the difference?
- Line 177: there seems to be an inconsistency: is it 500m2 grid or grid with a length of 500m?
- Line 196: if the origin and destination were different, how was "income" of the participant derived?
- Line 197: how did you infer the participant's address-origin or destination point?
- What are the 3 components in Table 3- please name them.
- Table 4: how is Greenway network density measured?
- Line 233: please explain how “weekend status” is used a control measure? By creating a dichotomous variable (i.e. 1,0)?
- Line 242: Were BE data averaged for each cell?
- Table 5: If Table 5 represents the hypothesis effect signs, then this should be clarified here as not "built" but "hypothesized".
Results
- Line 275: “The residential built environment in the clustered area was more accessible” - Not clear how Fig 6 shows this? It only shows the sport facility density.
- Table 8: need to report also the standardised B to be able to compare the contribution of each variable to the model.
- Line 313: “As shown in Table 8, the accessibility of the residential built environment in the clustered area was much higher than that in the sprawled area.” - not clear? Where in Table 8 shows this? Which accessibility measure are the authors referring to- street network density, bus stop density or facility density?
- Line 316: “there was no significant gap in the clustered area” - you mean "significant difference"?
- Line 351: “Overlapping the green lands, scenic spots and PA diversity (Figure 5)” - This figure does not show PA diversity.
- Line 363: “Accessibility is the fundamental and necessary condition” – 1) we do not know its relative contribution to the models, and 2) in Clustered areas, it was insignificant. So, I do not agree how this can be assumed?
- Lines 377-378: “the clustered structure successfully controls the scale of urban construction by natural and historical resources” – not clear? Please re-phrase and clarify.
Conclusions
- Line 412: “clustered urban structure should be advocated” – do the authors mean “compact urban structure”? The definition of clustered and sprawled areas and how they were measured should be clarified early on.
- The contribution of this study to the literature should be clarified.
Grammar
- There are also some grammatical issues which need to be reviewed. For example; “built environment also promote”- should be “promotes”; “those samples started”- should be “that started”
Round 2
Reviewer 2 Report
Thank you for the revisions.
Specific comments:
- Please change "discovered the relationships" to "analysed the relationships".
Author Response
Dear Reviewer,
As advised by the Editor in the email to us dated June 18, 2021, we would like to re-submit the paper (ijerph-1233440), “The effects of residential built environment on supporting physical activity diversity in high-density cities: a case study in Shenzhen, China” for the publication consideration in International Journal of Environmental Research and Public Health.
We would like to sincerely thank the reviewer for such valuable comments, which helped us improve the quality of our manuscript. We revised the manuscript carefully and tried to avoid any grammatical or syntax errors. The responses to the comment have been revised, appearing in blue font.
Response to the comment:
- Please change "discovered the relationships" to "analysed the relationships".
Response: Thank you for pointing this to us, we changed the "discovered the relationships" to "relationship analyzed" in the manuscript (See page 1, line 14).
We look forward to hearing from you in due time regarding our submission and to respond to any further questions and comments you may have.
Best regards,
The Authors
Reviewer 4 Report
I thank the authors for the revisions. The paper is now ready to be published.
Author Response
Dear Reviewer,
As advised by the Editor in the email to us dated June 18, 2021, we would like to re-submit the paper (ijerph-1233440), “The effects of residential built environment on supporting physical activity diversity in high-density cities: a case study in Shenzhen, China” for the publication consideration in International Journal of Environmental Research and Public Health.
We would like to sincerely thank your affirmation of our research, and we greatly appreciate your careful consideration and comments, which helped us revise our manuscript and improved the quality of the paper.
We look forward to hearing from you in due time regarding our submission and to respond to any further questions and comments you may have.
Best regards,
The Authors